# Systematic Literature Review of Swarm Robotics Strategies Applied to Target Search Problem with Environment Constraints

Zool Hilmi Ismail [1,2,*] and Mohd Ghazali Mohd Hamami [2,3]

1   Centre for Artificial Intelligence and Robotics, Universiti Teknologi Malaysia, Jalan Sultan Yahya Petra, Kuala Lumpur 54100, Malaysia
2   Malaysia-Japan International Institute of Technology, Universiti Teknologi Malaysia, Jalan Sultan Yahya Petra, Kuala Lumpur 54100, Malaysia; ghazali.hamami@uitm.edu.my
3   Faculty of Mechanical Engineering, Universiti Teknologi MARA Cawangan Johor, Kampus Pasir Gudang, Jalan Purnama, Masai 81750, Malaysia
*   Correspondence: zool@utm.my; Tel.: +60-199816001

**Featured Application: In this work, a systematic literature review of SR strategies has been applied to target search problems with environment constraints, to show which are being explored in the fields as well as the current state-of-the-art SR approaches performance which is delivered.**

**Abstract:** Target searching is a well-known but difficult problem in many research domains, including computational intelligence, swarm intelligence, and robotics. The main goal is to search for the targets within the specific boundary with the minimum time that is required and the obstacle avoidance that has been equipped in place. Swarm robotics (SR) is an extension of the multi-robot system that particularly discovers a concept of coordination, collaboration, and communication among a large number of robots. Because the robots are collaborating and working together, the task that is given will be completed faster compared to using a single robot. Thus, searching for single or multiple targets with swarm robots is a significant and realistic approach. Robustness, flexibility, and scalability, which are supported by distributed sensing, also make the swarm robots strategy suitable for target searching problems in real-world applications. The purpose of this article is to deliver a systematic literature review of SR strategies that are applied to target search problems, so as to show which are being explored in the fields as well as the performance of current state-of-the-art SR approaches. This review extracts data from four scientific databases and filters with two established high-indexed databases (Scopus and Web of Science). Notably, 25 selected articles fell under two main categories in environment complexity, namely empty space and cluttered. There are four strategies which have been compiled for both empty space and cluttered categories, namely, bio-inspired mechanism, behavior-based mechanism, random strategy mechanism, and hybrid mechanism.

**Keywords:** artificial intelligence; swarm intelligence; swarm robotics; systematic literature review; target search

## 1. Introduction

In the past 50 years, artificial intelligence (AI) has been centered on developing computers and algorithms that focus on human cognitive abilities. This type of AI, which is also called mainstream AI, has proven to be very successful to solve problems that are computationally heavy and lack human efficiency, such as controlling power plant systems or aircraft dynamics. In the mid-1980s, there was a surge to explore a new type of AI inspired by biological intelligence. It is called bio-inspired AI, the 21st century artificial intelligence (Figure 1). Accordingly, the bio-inspired AI can be categorized into seven sub-categories, which are evolutionary systems, cellular systems, neural systems, development

systems, immune systems, behavioral systems, and collective systems [1]. One form of bio-inspired AI which has been emerging rapidly is swarm intelligence (SI).

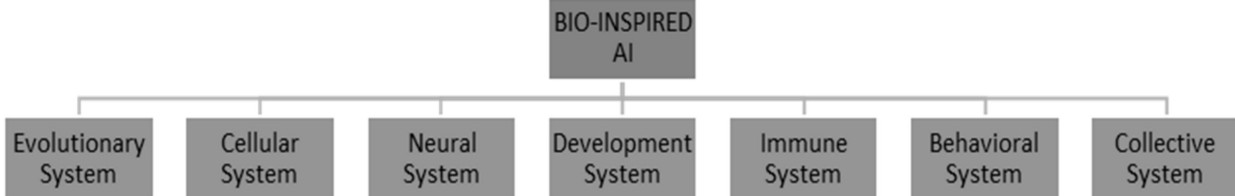

**Figure 1.** Bio-inspired artificial intelligence categorization.

SI is inspired by studying the behavior of groups, or swarms, of a biological organism. Social insects such as ants, bees, and termites live in colonies or swarms, and every single member in the colony seems to have its purpose yet the colony (swarm) appears organized [2]. The most unique features of such swarms are the fact that they portray complex collective behavior despite the simplicity of the individuals (agents) that make up the swarm. The common complex collective behaviors are aggregation [3], foraging [4], flocking [5], cooperation [6], and stigmergy [7]. Models of this system have been proven to solve a difficult and complex real-world problem such as optimization [8,9], and target search [10,11]. The main representatives of SI approaches are particle swarm optimization (PSO) [12], bees algorithm (BA) [13], artificial bee colony optimization (ABC) [14], ant colony optimization (ACO) [15,16], bacterial foraging optimization (BFO) [4], glowworm swarm optimization (GSO) [17], and firefly algorithm (FA) [18]. SI has also been actively incorporated with the latest robotic technology by researchers and technologists, which has emerged as the swarm robotics (SR) domain. SR, together with the implementation of SI methods, represents the computational agent into physically implemented or simulated robotic devices. SR strategies have continuously increased the attention in many applications, especially those that are 3D (dangerous, dirty, and dull) mission-related, such as search and rescue [19], pollution detection [20], and natural disaster monitoring [21]. These types of applications require a large number of agents, are time consuming and may even be dangerous to a human being [22]. In addition, all the given applications have a common important task, which is to cooperatively search for targets in unknown environments. Compared to a single robot, swarm robots can significantly improve efficiency and provide better robustness and adaptability in target searching tasks [23].

The problem of target searching has continued for a very long time, and more civilian applications have emerged. This includes a wide variety of high-impact applications; for example, rescue operations in disaster areas, exploration for natural resources, environmental monitoring, and air surveillance.

By adapting the SR strategies, swarm robots can sense the environment, estimate the target's distance, and cooperatively coordinate swarm movement by taking advantage of this information. The ultimate goal is to search for targets within the minimum time while avoiding collisions with other robots [11]. Therefore, the purpose of this paper is to deliver a systematic literature review (SLR) covering the main published solutions of target search problems using SR strategies. This paper expands the algorithm behind the SR strategies and their significant result, challenges, and future dimensions towards the target search problem.

The organization of this paper is structured as follows: Section 2 describes the planning and execution of the SLR; Section 3 presents an overview and characteristics of SR; Section 4 gives a summary of the studied literature, delivers the answers to the research questions, and the explanation of the main characteristic of SR strategies that are taken in the target search problem; and finally, in Section 5, the paper's contributions are presented, and concluding remarks are summarized.

## 2. Systematic Literature Review

A systematic literature review, or SLR, is a popular technique which is widely used to distinguish, evaluate, and interpret relevant parts of research for a specific issue, area, or phenomenon of interest [24]. A systematic review is carried out with a formulated question that uses systematic and explicit methods to identify, select, and critically appraise relevant research and to collect and analyze data from studies that are included in the review. The approached methodology by Kitchenham in [24] has been used as a guideline for this SLR implementation.

### 2.1. Literature Review Planning Protocol

This paper considers the following planning protocol for the review:

1.  Research Questions

    $Q_1$. What SR strategies are being used to perform the target search task?
    $Q_2$. How many targets have been searched?
    $Q_3$. What is the target versus search agent ratio?
    $Q_4$. What is the mobility of the targets?
    $Q_5$. What is the state of the environment's complexity?
    $Q_6$. Is the strategy verified by simulation only or both simulation and the real robot platform?
    $Q_7$. How are the SR strategies implemented in the target search problem?

2.  Databases for Literature Search

    This systematic study was carried out on four well-established literature databases with scientific scope, which are Springer, IEEE Xplore Digital Library, ScienceDirect, and MDPI.

3.  Exclusion Criteria

    $E_1$. Works that are not included either in the Scopus or Web of Science (WOS) database.
    $E_2$. Works that are not related to the target search problem and SR.
    $E_3$. Works that do not present any type of experimentation or comparison result and only make propositions.

4.  Quality Criterion

    $QC_1$. Papers that compare the target search problem result using different SR strategies.

5.  Data Extraction Fields

    $D_1$. Implemented SR strategy, being able to consider any SR strategies from classical to new and state-of-the-art strategies.
    $D_2$. Target quantity that searches either a single target or multiple targets.
    $D_3$. Mobility of the target; either static or dynamic.
    $D_4$. Target versus search agent ratio data.
    $D_5$. Environment complexity that states either an empty space state or a cluttered state.
    $D_6$. Verification of the strategies either using simulation only or both simulation and the real robot platform.
    $D_7$. Inspiration phenomenon and mechanism of SR strategies that have been implemented in the target search problem.

### 2.2. Execution

The choice of keywords for building the search strings was based on terms that were commonly found in the literature and the term that was related to this review (i.e., swarm robotics methods that were applied to target the search problem). For the SLR execution, specific keyword strings were formulated and used for each database (Springer, IEEE Xplore, ScienceDirect, and MDPI). The details are listed below:

(1) IEEE Xplore: ("Swarm robot*" AND "target search*") with, eta-data in the command search;

(2) Springer: ("Swarm robot*" AND "target search*");

(3) ScienceDirect: ("Swarm robot" AND "target search");

(4) MDPI: ("Swarm robot*" AND "target search*").

The survey took place in the week ending 31 January 2021. Figure 2 portrays the number of searched documents in the databases using the selected keyword strings. The total number of searched papers was 75; however, only 29 of the papers were enlisted in Scopus and Web of Science, and as such 46 out of the 75 papers were excluded as per the exclusion criteria $E_1$. Out of these 29 papers, four articles were rejected using the exclusion criteria $E_2$ and $E_3$. Thus, the total number of papers that were selected for this review was 25 articles.

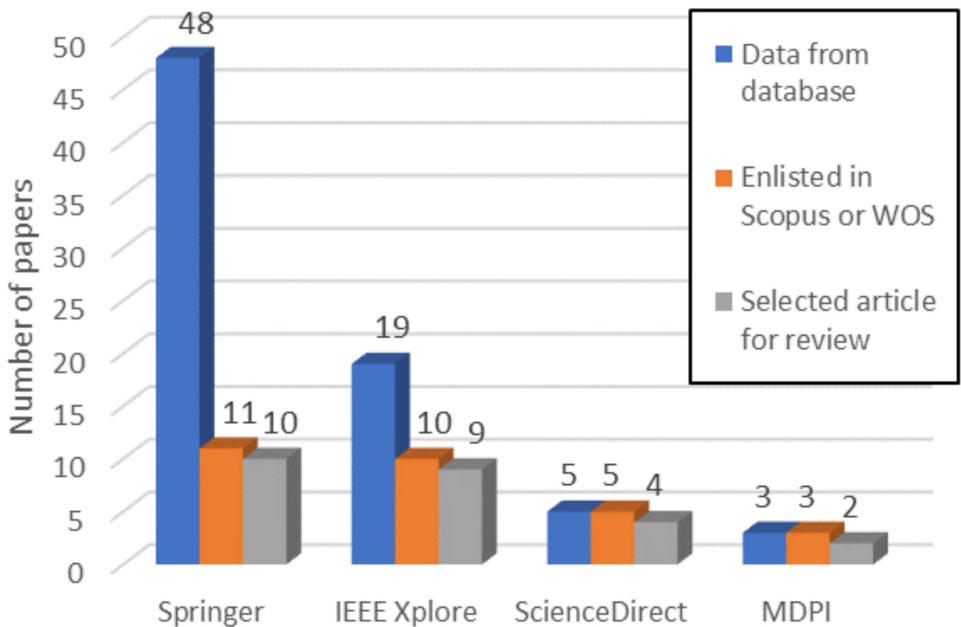

**Figure 2.** Number of papers in the databases using the extraction criteria.

## 3. Background on Swarm Robotics

It is essential to introduce some background on SR strategy before presenting the review analysis design. SR is an expanding field that is inspired by the natural and self-organized behaviors of social animals such as a bird flock or an ant colony [25]. These inspirations from the natural behavior of social animals can be applied with the principles of self-organization to collections of simple, autonomous robots [26]. The robots must not have a sophisticated and complicated system and do not use a complex algorithm. The main idea of SR is to gain the advantage of simple interactions among the robots in order to solve complex problems using emergent behavior, similar to how social insects operate [27].

To achieve this idealized simplicity, SR strategy needs to have three main characteristics, i.e., robustness, flexibility, and scalability [28]. The first characteristic, robustness, enables the swarm to continue to function as a system without impacting the performance even if several agents fail to function. The second characteristic is flexibility, which enables the swarm system to adapt and promptly change to the dynamic environment or the changes in task difficulty. The third characteristic is scalability, which enables the system to operate with small or large numbers of agents without affecting the system performance and efficiency.

To achieve the above SR characteristics, the SR strategy needs to be designed with a set of criteria [29]:

1. The robot swarm must be made up of a number of autonomous robots with the ability to sense and actuate in a real environment;

2.  The number of robots in the swarm must be large, at least as large as the control rules authorize;
3.  Robots must be homogenous. There can be different types of robots in the swarm, however, not too varied;
4.  The robots must not be unable or be inefficient in the main task that they must solve, i.e., they need to cooperate in order to succeed or to improve the performance;
5.  Robots are limited to local communication and sensing capabilities; this is to ensure the coordination is in distribution mode, so that scalability will become one of the properties of the system.

For further literature review about the characteristics and main properties of SR strategy, the research by Navarro and Matía [30] is recommended.

## 4. Result of the Systematic Literature Review

### 4.1. Publication Distribution over the Years

Figure 3 shows the number of articles that have been published from 2005 until 2020 (based on the extraction criteria in Section 2) in five-year groups. This chronological distribution with the exponential trend is evidence that interests in SR strategies have been growing and have gained attention to solving the target search problem. The papers were published in well-established, high-indexed databases, which began with only a single paper that was published between 2005 to 2010, and five papers between 2011 and 2015; the trend of publishing in the high-indexed databases continued from 2016 until 2020, which accounted for the highest number of published papers—19 altogether, with an average of 3.8 papers per year. The analysis results reveal that there has been a substantial increase in the number of literature articles since 2016. This trend may be supported by the incremental expansion of SI throughout the years with the support of the increasing computational intelligence performance [31]. This would reveal a promising established research area in SR strategies for the target search problem.

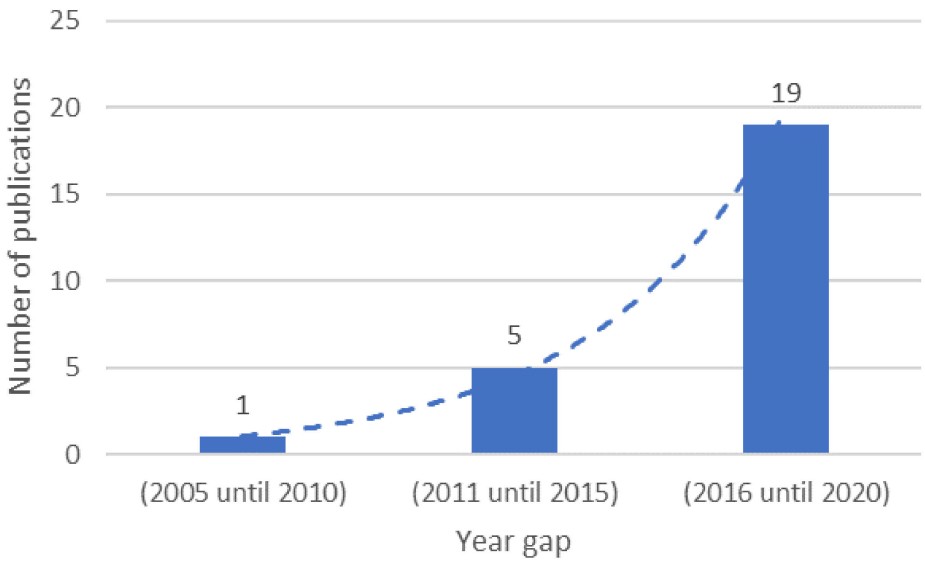

**Figure 3.** The number of papers per year with a trend line.

### 4.2. Publication Distribution among Journals and Conferences

The selected papers that were based on SR strategies, applied to target search problems, have been published in a wide range and variety of journals, as well as conferences, which mainly focused on the engineering fields. From the 25 selected papers, 14 were published in journals, and the remaining 11 papers were published in conferences.

There were 14 research papers that had been published across nine journals: this was led by *Lecture Notes in Computer Science* (including subseries *Lecture Notes in Artificial*

*Intelligence* and *Lecture Notes in Bioinformatics*) with four publications; the *Journal of Robotics and Autonomous Systems* and the journal *Sensors*, each with two publications; and the remaining six publications were each published in *Studies in Computational Intelligence*, the *Iranian Journal of Science and Technology—Transaction of Electrical Engineering, IEEE/ACM Transaction on Computational Biology Bioinformatics, IEEE Access, Applied Soft Computing Journal*, and *Neurocomputing* (Figure 4). Most of the journals covered the application of artificial intelligence and computational intelligence in the engineering domains.

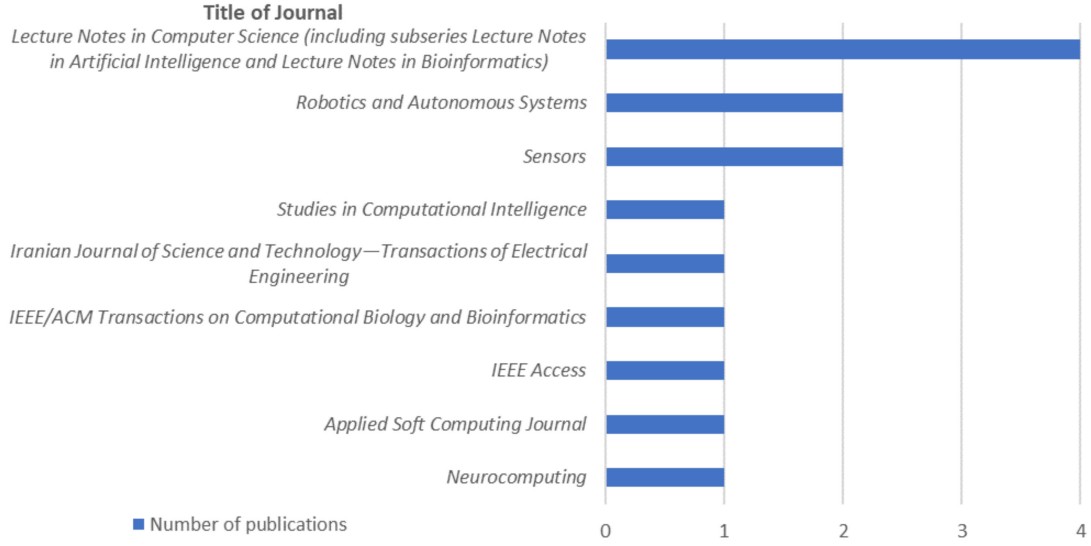

**Figure 4.** Number of publications per journal.

On the other hand, the research papers belonged to nine conferences. Both the Advances in Swarm Intelligence (ICSI) and IEEE Congress on Evolutionary Computation (CEC) were the top conferences, with two publications each (Figure 5). Each of the remaining conference papers were published in seven conferences which covered a wide range of domains; mostly the focus was on AI, computational intelligence, and controlling intelligent systems. It should be highlighted that only Information and Communication Technology (ICICT) focuses specifically on the communication and information domain, which is different from other conference domain focuses.

*4.3. Citation Analysis*

One of the quality measures of the published paper is how many times the article has been cited by other researchers. To carry out a citation analysis and to maintain the quality of the selected paper, the Scopus and WOS platform were selected to determine the number of citations of selected papers in the review. Table 1 portrays the citation quantity for all the selected papers in a decreasing order manner. The research paper entitled "Self-organized Swarm Robots for Target Search and Trapping Inspired by Bacterial Chemotaxis" [32] outperformed others, with the highest recorded number of citations (33). The second-highest recorded number of citations, with 17, was the research paper entitled "Group Explosion Strategy for Searching Multiple Target using SR" [22]. The third-highest citation is by the SR community, which is the paper that has been published by Tang et al. [33], with eight citations. The paper discussed the stigmergy strategy approach towards the target search problem. From the 25 selected papers, 12 papers have not been cited even once, and thus they have been excluded from Table 1. Based on the citation analysis results, the average number of citations of all the selected research papers was 3.6.

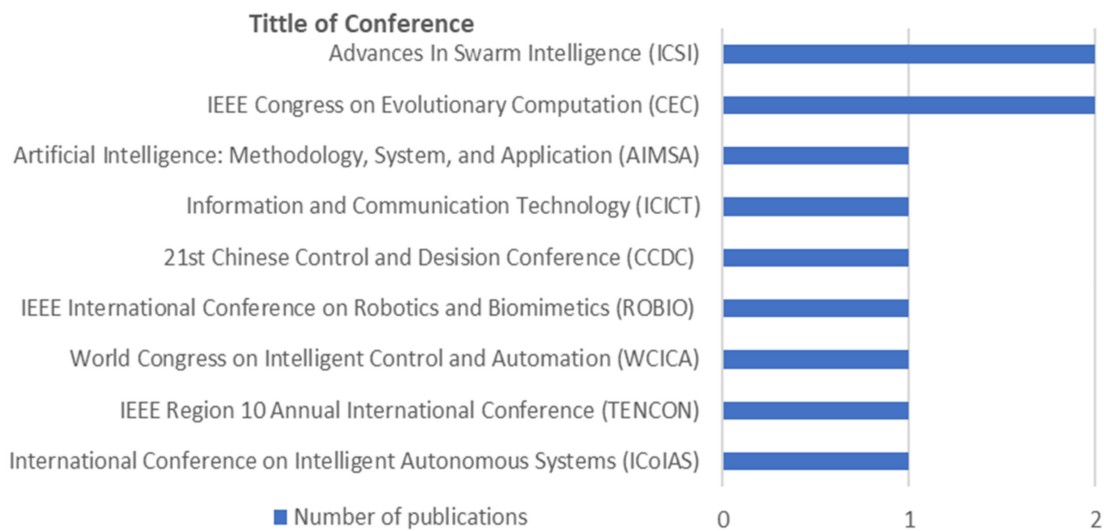

**Figure 5.** Number of publications per conference.

**Table 1.** Summary of the top 13 cited articles.

| Title | Publication Year | Citations |
| --- | --- | --- |
| Self-organized swarm robots for target search and trapping inspired by Bacterial Chemotaxis [32] | 2015 | 33 |
| Group explosion strategy for searching multiple targets using swarm robotic [22] | 2013 | 17 |
| A stigmergy based search method for swarm robots [33] | 2017 | 8 |
| The multi-target search problem with environmental restrictions in swarm robotics [34] | 2014 | 7 |
| Avoiding decoys in multiple targets searching problems using swarm robotics [35] | 2014 | 6 |
| Swarm robots search for multiple targets based on an improved grouping strategy [36] | 2018 | 4 |
| Target searching and trapping for swarm robots with modified bacterial foraging optimization algorithm [37] | 2015 | 4 |
| Comparison of a real Kilobot robot implementation with its computer simulation focusing on target-searching algorithms [38] | 2018 | 3 |
| A comparative study of biology-inspired algorithms applied to swarm robots target searching [39] | 2016 | 2 |
| Triangle formation based multiple targets search using a swarm of robots [40] | 2016 | 2 |
| Target search using swarm robots with kinematic constraints [41] | 2009 | 2 |
| Optimal tree search by a swarm of mobile robots [42] | 2018 | 1 |
| A grouping method for multiple targets search using swarm robots [43] | 2016 | 1 |

*4.4. Research Strategy Analysis*

After an analysis of the 25 selected papers that were published from 2009 until 2020, Table 2 was constructed based on the extraction criteria in Section 2; it summarizes and highlights all the important elements of the most recent high-impact papers for target search problem with environment constraints using the SR strategy. The first column lists the paper's reference; the second column is the proposed method(s), recording the implemented SR strategy in the paper; the third column states the number of targets that are available, either single or multiple; the fourth column quantifies the ratio between targets and agents; the fifth column records the targets which are stated whether in the

static or dynamic condition; the sixth column specifies the environment complexity in the simulation or experimental setup, which is either empty space or cluttered; the seventh column mentions the algorithm verification selection, either the simulation only or both simulation and real robot platform; the last column contains the natural phenomenon inspiration that has been used in the SR strategy.

**Table 2.** A summary of the most recent papers for target search problem with environment constraint.

| Reference | SR Proposed Method | Number of Target(s) | Targets/Agents Ratio | Mobility of Targets | Environment Complexity | Verification Method | Inspiration Phenomenon |
|---|---|---|---|---|---|---|---|
| [41] | Kinematic constrain target search strategy | Single | <1 | Static | Empty space | Simulation | Flocking behavior of birds |
| [22] | Group explosion strategy (GES) | Multiple | >1 | Static | Empty space | Simulation | Explosion phenomenon in nature |
| [34] | Three restriction-handling strategies | Multiple | <1 | Static | Cluttered | Simulation | Explosion phenomenon in nature and flocking behavior of birds |
| [35] | Decoy-avoiding strategies | Multiple | <1 | Static | Cluttered | Simulation | Explosion phenomenon in nature and flocking behavior of birds |
| [37] | Modified bacterial foraging optimization (MBFO) strategy | Multiple | <1 | Static | Cluttered | Simulation | Foraging behavior of bacteria (*Escherichia coli*) |
| [32] | Self-organized target search and trapping strategy | Multiple | <1 | Dynamic | Empty space | Simulation | Foraging behavior of bacteria (*Escherichia coli*) |
| [43] | Integrated strategy based on a modified particle swarm optimization (PSO) algorithm and a grouping strategy | Multiple | <1 | Static | Empty space | Simulation | Flocking behavior of birds |
| [39] | Comparative between PSO, ant colony optimization (ACO) and genetic algorithm (GA) in target search | Single | <1 | Static | Cluttered | Simulation | Flocking behavior of birds, foraging behavior of ants, and natural selection by biologically inspired selection |
| [40] | Triangle formation search (TFS) strategy | Multiple | <1 | Static | Empty space | Simulation | Triangle formation behavior |
| [44] | Sweep cleaning protocol strategy | Single | <1 | Dynamic | Empty space | Simulation | Sweep cleaning behavior |
| [33] | Finite-state machine and coding phase pheromone strategy | Single | <1 | Static | Empty space | Simulation and real robot | Cockroach behavior |
| [45] | Multi-objective particle swarm optimization (MOPSO) and energy-saving decision rules strategy | Single | <1 | Static | Empty space | Simulation | Flocking behavior of birds |
| [46] | Flying ant-like search strategy | Multiple | <1 | Static | Cluttered | Simulation | Flying ant behavior |
| [36] | Improved grouping strategy based on constriction factor particle swarm optimization (CFPSO) | Multiple | <1 | Static | Cluttered | Simulation | Flocking behavior of birds |
| [47] | Dispersal search strategy | Single | <1 | Static | Empty space | Simulation and real robot | Random walk behavior |
| [42] | Tree search strategy | Single | <1 | Static | Cluttered | Simulation | Tree target searching behavior |
| [48] | Lévy walk strategy | Multiple | <1 | Static | Empty space | Simulation | Lévy flight behavior |
| [49] | Chemotaxis behavior strategy | Single | <1 | Dynamic | Cluttered | Simulation and real robot | Microorganism behavior |
| [50] | A probabilistic finite state machine-based strategy | Multiple | <1 | Static | Empty space | Simulation | Random walk and triangle estimation technology |
| [51] | A two-stage imitation learning framework strategy | Multiple | <1 | Static | Empty space | Simulation | Deep learning and evolutionary algorithm. |
| [52] | Dynamic target searching and tracking stigmergy strategy | Single | <1 | Dynamic | Empty space | Simulation and real robot | Foraging behavior of ant |
| [53] | A pheromone underwater robot monitoring strategy | Multiple | <1 | Static | Empty space | Simulation and real robot | Foraging behavior of ant |
| [54] | Repulsion-based robotic Darwinian particle swarm optimization (RDPSO) strategy | Single | <1 | Static | Cluttered | Simulation | Flocking behavior of birds |
| [55] | Bean optimization-based cooperation strategy | Single | <1 | Static | Empty space | Simulation | Natural plant evolution behavior |
| [11] | A distributed strategy for multi-target search in an unknown environment strategy | Multiple | <1 | Static | Empty space | Simulation | Pedestrian behavior |

Analysis of the review was carried out, and the result verified that 64% (16/25) of the proposed methods used the SI approach in their SR strategies. Particle swarm optimization (PSO) was the most preferred SI algorithm, with 9/16 research papers having implemented it. PSO was originally suitable for solving global optimization problems, and target search problems had a similar characteristic of global optimization problems especially in multi-target search problems. Six of the papers chose the behavior-based approach in their SR strategies and the remaining three papers implemented the random walk method in their SR strategies to tackle the target search problem.

Data from Table 2 also shows that 56% (14/25) of the papers set their experiments with multiple targets over single targets, and in 96% (24/25) of papers, the search agents were more than an available target. In terms of the target's mobility, 84% (21/25) of the papers set their targets in static conditions. These results show that the researchers focused on emerging swarm behaviors and interactions between the search agents during the target search problem.

Review analysis also revealed that 64% (16/25) of research papers set the search environment space without any obstacles (empty space). The final data that were disclosed in Table 2 was the algorithm verification method. A total of 80% (20/25) of the papers only used a simulation platform to verify their proposed algorithm. There are two reasons why the simulation platform is popular among researchers: firstly, the possibility to test a vast number of agents without investing a large budget; second is the time-consumption matter.

Through the analysis of the above characteristics, the research questions that had been mentioned in Section 2 were answered. The next subsection describes the main characteristic of the most used SR strategies and how they are implemented on the target search problem.

### 4.4.1. Particle Swarm Optimization

Particle swarm optimization, or PSO, took inspiration from birds' flocking behavior and was introduced by Kennedy and Eberhart [56]. Agents that are considered as particles in PSO are flown through a problem space to consider the best result (fitness) position within the search space. During the searching process, the particles update their velocity and latest best position as well as the overall best position that has been achieved within the neighborhood, either a global optimum or local optimum. For each $i$th iteration, the velocity ($v$) and position ($x$) will be updated as the particle performs the $k$th step of the PSO algorithm:

$$v_i^k = \omega v_i^{k-1} + c_1 r_1 \left( p_i - x_i^{k-1} \right) + c_2 r_2 \left( p_n - x_i^{k-1} \right) \tag{1}$$

And:

$$x_i^k = x_i^{k-1} + v_i^k \tag{2}$$

where $v_i^k$ and $x_i^k$ are designated as velocity and position vectors for the $i$th particle in the $k$th time-step, $p_i$ represents the personal best position of the $i$th particle, while $p_n$ represents the overall best position of all particles within the neighborhood. $\omega$ is the inertia weight that is introduced to balance between the global search and local search by controlling how much the current velocity of the particle contributes to its velocity in the next iteration [57]. $c_1$ and $c_2$ are referred to as cognitive scaling and social scaling factors, respectively, while $r_1$ and $r_2$ are random numbers that are drawn from a uniform distribution. The PSO algorithm was originally developed for solving global optimization problems, and there is a similarity between the objective of multiple target problems and global optimization problems, which is the searching for the best solution with minimal time-consumption. This is the main reason why the PSO method has been adapted by the researcher community when handling the target search problem using SR strategies.

Songdong et al. in [41] reported an approach in controlling swarm autonomous wheeled mobile robots with non-holonomic constraints to search for a single target. The proposed SR strategy focused on kinematic constrain target search strategy. The proposed algorithm compared both similarities and differences with the PSO, because the proposed

algorithm was inspired by the PSO algorithm. The simulation results demonstrated that the proposed algorithm had successfully searched the target and portrayed the swarm robots to be able to cooperatively work together during the target search process.

An integrated method based on a modified PSO algorithm and a grouping strategy for multiple-target search swarm robots was developed by Tang et al. in [43]. The developed algorithm has been evaluated in the simulation platform and was compared with the basic PSO algorithm to show the algorithm's effectiveness. Based on the algorithm comparison, the proposed method portrayed good adaptability and a high success rate in searching for multiple targets. In 2017, Tang et al. [36] has continuously improved the method by incorporating the obstacle avoidance behavior into the algorithm. By considering the obstacle avoidance behavior in the algorithm, the improved grouping strategy (IGES) for multiple-target search swarm demonstrates the significant improvement in terms of adaptability, accuracy, and efficiency during the multiple target search process.

Previous work by Yuen et al. [45] proposed an algorithm of swarm robots dependent solely on light energy to survive and complete target searching tasks in an unknown area. The research not only considered target searching, but also energy consumption matters. To increase the complication of the target search problem, the target and energy charging points were also scattered around the search area. For energy saving decision rules, the multi-objective particle swarm optimization (MOPSO) has been considered. Several sets of simulation experiments were conducted.

The results showed an optimum quantity of 15 robot swarm systems that were able to search a single target and stabilize the energy level simultaneously for the long-term. This proved that the energy-optimized MOPSO as a design framework is suitable for a long-term target searching swarm system.

A novel algorithm, repulsion-based robotic Darwinian particle swarm optimization (Rb-RDPSO) that combines the Darwinian principle of "survival of the fittest" with an ion-based repulsion mechanism has been introduced to tackle the target search problem [54]. Several simulations in a cluttered environment have been designed to validate the proposed method. The proposed method is inspired by PSO and has been compared with the traditional robotic Darwinian particle swarm optimization (RDPSO), the robotic PSO, and the distributed PSO. The outcomes portray the advantages of the proposed method efficiency in both speed and search results. The method also showed a superior performance compared to other strategies as the number of swarm agents decreased.

### 4.4.2. Behavior-Based Approach

The idea of behavior-based SR is inspired by the implementation of behavior-based AI theory using an architecture called the "subsumption architecture" by [1]. The robots' behaviors "subsume" each other depending on the results from a variety of inputs, such as vision and pressure sensor data. At any given time, only one behavior will be activated. The active behavior varies based on the gained sensor data. One of the earliest works of behavior-based formation control was applied to groups of robots by Balch and Arkin [58]. Their work successfully integrated formation behaviors with navigation and obstacle avoidance both in simulation and on a set of land-based ground vehicles. The behavior-based concept has continuously been developed throughout the years and has been implemented in the SR fields.

Zheng and Tan in [22] proposed a swarm robot searching strategy of multiple targets in obstructive environments. The searching strategy was named as group explosion strategy (GES) and has been inspired by the explosion phenomenon behavior in nature. Each swarm agent is self-adaptively divided into small groups and performs the target searching process independently. Through its advantages of quick convergence from intra-group cooperation and capability of multiple targets searching in parallel from intergroup cooperation, GES simulation outputs portray a great efficiency in energy consumption and target searching time. The proposed strategy also shows great stability in obstructive environments.

Research work performed by Li and Tan in [34] gives attention to the basic search problem of multi-target search in several conditions of environmental restrictions. They proposed the IGES to solve the multi-target search problem. The strategy was an improved version of the previous GES [22] and was inspired by a firework explosion phenomenon behavior with three restriction-handling strategies. The proposed IGES strategy has been tested in the simulation and compared with the GES and RPSO methods. The simulation outcomes show that IGES is more efficient and has great stability during the searching process compares to GES and RPSO methods. The strategy also has greater compatibility with other restriction-handling strategies.

A simple cooperative strategy as the baseline of target searching problems with a new type of object (decoys) was proposed by Zheng et al. [35] and was also inspired by the firework explosion phenomenon behavior. In the paper, decoys were also considered as targets that can be sensed but cannot be collected by the robots. This becomes a problem that needs to be considered in the multi-target search task. The method introduced a simple cooperative strategy to solve the problem by comparing it with a non-cooperative strategy as the baseline. The strategies can be integrated with other searching algorithms and provide a solution for avoiding decoys. Simulation results portray that the strategy has almost similar computation overload compared to other strategies, however has better performance in iterations and visiting times of decoys. The strategy shows great adaptiveness to large-scale problems and functions more efficiently when the quantity of decoys or robots increases.

The implementation of a "sweep cleaning" protocol in a swarm of flying robots has been carried out by Fermin et al. [44]. The research implements an idea from sweep cleaning behavior. The "dynamic cleaners" problem is one of the attractive applications proposed by the research community in the SI field. The swarm converges in the given area and searches the target area where the contamination spreads. Based on the simulation outcomes, the accuracy of the introduced strategy is 89.8%, and the cleaning time process decreases by 12.87% for each single agent incrementation. The strategy can be implemented in several real-life world applications such as target searching and search and rescue operations.

Researchers also investigate the use of multi-modal locomotion on a swarm of robots through a multi-target search algorithm that is inspired by flying ants' behaviors [46]. The proposed strategy focuses on the SI elements such as distributivity, robustness, and scalability to guaranty that efficient exploration is archived. The simulation outputs portray that efficient exploration was achieved at the macro level of the swarm robots during the multi-target searching process.

A stigmergy-based search method [33] has been presented for swarm robots. The proposed strategy implements the finite-state machine and coding phase pheromone strategy inspired by cockroach behavior. It utilized a pheromone technique by arranging radio frequency identification (RFID) tags in the environment as a carrier of the pheromone. Through several numerical experiments and verifications, the results proved the applicability of the proposed algorithm.

In the target searching task, all the agents in the swarm were tasked with gathering at the specifically designated node, which was termed as the target node. The scope of the solution increases during target search if the graph is guaranteed to be explored completely. In conjunction with the requirements, Sinha and Mukhopadhyaya in [42] proposed a target search algorithm of limited visibility swarm of asynchronous robots based on tree search behavior. Each node of the tree was assumed to be attached with memory. The target node was initially visible to at least a single agent in the swarm. The algorithm takes computational cycles to gather all the agents at the target node after the exploration of the tree has been completed. The simulation outcomes show the efficiency of the proposed tree search strategy in the target search problem.

### 4.4.3. Random Walk or RW

An environment or target can be searched more effectively and efficiently if a suitable search strategy is used. Due to the individual ability limitation of swarm robots, such as local sensing and low processing power, random searching is another preferred approach that has been selected by researchers in the SR domain. The commonly used random walk methods are Brownian motion (BM) and Lévy flight (LF), which both mimic and are inspired by the self-organized behavior of social insects [59]. BM is the random movement of particles that are suspended in a fluid, resulted from fast particles colliding between the molecules of the fluid [60]. LF is an RW by which the agents can travel a significantly large distance by taking many short steps and the occasional long step [61]. This is possible due to the step size that has a power–law distribution, and agents that use LF are more likely to reach the furthest area fasters than those which use BM. Each LF step orientation is sampled from a uniform distribution, while the step lengths are sampled from a heavy-tail (power–law) distribution:

$$p(l) \sim 1^{-(\alpha+1)} \tag{3}$$

Previous work by Zhong et al. [38] presents the implementation of a targeted search by SR algorithm using a dispersal search strategy that is based on the random walk method. The developed strategy has been tested in the Virtual Robot Experimental Platform (V-REP) simulator and was then validated by the Kilobots robot platform. The algorithm has three main steps, which are dispersal target searching, obstacle avoidance, and target surrounding. Both results in the simulation and robot platforms indicate the functionality and quality of the proposed strategy in the target search problem.

A new target search behavioral algorithm that preserves Lévy properties at the collective level in the unknown environment under limited energy and deadline conditions has been proposed by Khaluf et al. [48]. The paper highlights the problem of how Lévy properties can disappear in larger robot swarm sizes due to spatial interferences. The simulation results define the algorithm that can accelerate target search processes in large unknown environments by parallelizing Lévy exploration.

### 4.4.4. Hybrid Strategy

Hybrid strategy that incorporates two or more techniques may compensate for the vulnerability of one technique by making use of the other. Researchers often give special attention to this hybrid strategy solution. This approach can be seen in the triangle formation search (TFS) strategy for swarm robots and has been introduced by Li et al. [40] to overcome the multiple target search problem. The strategy is based on triangle formation and random search, focusing on balancing between exploration and exploitation during the target searching process. In addition, a new random walk strategy of linear ballistic motion, incorporated with triangle estimation, has been compared with the TFS strategy and the performance of a new random walk strategy shows its advantages and can serve as a benchmark in the multiple-target search problems.

### 4.4.5. Comparison of SR Strategies Applied to Target Search Problems

Table 3 portrays the comparison of SR strategies that have been implemented in target search problems. Particle swarm optimization (PSO) is well established in the target search problem domain as a result of its algorithm implementation simplicity. The PSO is also well-established for the global optimization problem where there is a similarity with the target searching problem, which is searching for the best solution in minimal time. However, the PSO strategy has a limitation in terms of its tendency to be trapped in a local minimum.

Table 3. Comparison of SR strategies applied to target search problems.

| SR Strategies | Mechanism | Advantages | Limitation |
|---|---|---|---|
| Particle Swarm Optimization | 1. Inspired from bird flocking behavior [12].<br>2. Particles (agents) flown through a problem space to search the best result (fitness) position within the search space.<br>3. Particle velocity, best position, and overall (global) best position will be updated during the search process | 1. Search is driven by social interaction among particles by consideration of each particles best position during the selection of the overall swarm position.<br>2. Simplicity of algorithm implementation.<br>3. Low in computational intensity. | 1. Has tendency to trap in a local minimum.<br>2. Does not have guaranteed convergence to a local or global minimum. |
| Behavior-based | 1. Inspired by subsumption architecture [1].<br>2. Only one behavior can be activated at any given time and the robots' output behavior depends on the activated behavior.<br>3. The active behavior varies based on gained sensor data. | The flexibility of its input sensor combination enable the behavior-based strategy to comply with several search environment scenarios. | 1. Requires sophisticated rule comparison analysis to solve large-scale and practical problems.<br>2. Takes a long computational time. |
| Random Walk | 1. Inspired by the self-organized behavior of social insects [50].<br>2. Most commonly used random walks are Brownian motion (BM) and Lévy flight (LF). | 1. Does not required any initial knowledge of target distribution and environment details.<br>2. Simple and easy to implement towards SR. | The random walk distribution and its properties tend to be lost when the swarm size increases. |
| Hybrid Strategy | Combining two or more SR strategies to overcome the limitation of each strategy. | Dependent on the combined strategies that have been implemented. | Dependent on the combined strategies that have been implemented. |

The behavior-based strategy has continuously been implemented in the robotic domain because of its input sensor combination flexibility which can comply with various types of search environments. On the other hand, the behavior-based strategy tends to take a long computational time during the solving of large-scale and practical problems due to sophisticated rule comparison analysis.

The random walk strategy has also been applied to the target search problem by several researchers [38,48]. The RW strategy does not require any initial knowledge of target distribution and environment details which is its advantage, but its distribution and properties tend to decline when the swarm size increases.

Each strategy has its advantages and limitations. Some research has proposed the hybrid strategy in order to maintain the advantages and to overcome the limitations of each strategy.

## 5. Conclusions and Future Works

This article covered the main papers of swarm robotics strategies that were applied to target search problem by systematically answering the research questions that had been described in the literature review planning protocol. Through this meticulous process, it was possible to identify that each proposed strategy had addressed a specific constraint or restriction; hence, it became difficult to compare them directly to each other. The SR strategies emerged as a new method for dealing with target search problems in conjunction with the rise of AI, particularly in the SI field. This is supported by the increasing demand for SR field utilization in high-impact applications such as exploration for natural sources

or search and rescue, where the target search task is one of the important elements of the problem.

During this review, it was noted that most of the SR strategies were using the SI approach, particularly the PSO method. The PSO algorithm was developed to focus on solving global optimization problems. There is a similarity between the objective of target problems and global optimization problems, i.e., searching for the best solution with minimal time-consumption. Due to this similarity, the PSO method is easily adapted to SR strategies during the solving of the target search problem. All the SR strategies that have been applied in target search problems are inspired by natural behavior; this proves that there is knowledge that can be gained from nature.

From the environmental constraints' point of view, out of the 25 strategies of the selected articles, 16 strategies were implemented in empty space and 9 strategies were applied to the cluttered environment. The strategies focused on bio-inspired mechanisms, behavior-based mechanisms, random strategy mechanisms, and hybrid mechanisms. These results show that the researchers focused more on emerging swarm behaviors and interactions between the search agents than the swarm interaction between the environment during the target search problem. For further recommendations, both interactions (interaction between search agents and swarm interaction between the environment) are equally important and need to be considered.

Additionally, it should be acknowledged that SR strategies, such as PSO, behavior-based and random walk, have been successfully applied to solve the target search problem. However, there are still some aspects in SR strategy applications towards target search problems that need to be further investigated. Thus, recommendations for future research include:

1. Dynamic environment simulation testing with various kind of targets which have different outcome values [22,62]. Most of the articles only focused on a static environment;
2. Real robot platform experiments to validate the simulation results, thus minimizing the real-world application gap [36,37]. Due to research capital limitations, most of the articles only verified the algorithm in a simulation platform;
3. Derive a mathematical model of the swarm robot interactions and design a suitable controller that comes with a certain proof of convergence [34]. Most of the articles only manually designed the local behaviors, analyzing them by trial and error until the desired swarm behaviors were achieved.

**Author Contributions:** Z.H.I.: Conceptualization, Z.H.I.; methodology, Z.H.I. and M.G.M.H.; validation Z.H.I. and M.G.M.H.; investigation, Z.H.I. and M.G.M.H.; visualization, Z.H.I. and M.G.M.H.; supervision, Z.H.I.; data curation, Z.H.I. and M.G.M.H.; writing—original draft preparation Z.H.I. and M.G.M.H.; writing—reviewing and editing, Z.H.I. and M.G.M.H. Both authors have read and agreed to the published version of the manuscript.

**Funding:** This work was supported in part by the Ministry of Higher Education, Malaysia, and AUN/SEED-Net under Grant no. R.K130000.7843.5F348 and R.K130000.7343.4B617.

**Institutional Review Board Statement:** Not applicable.

**Informed Consent Statement:** Not applicable.

**Data Availability Statement:** Not applicable.

**Acknowledgments:** Thanks to Universiti Teknologi Malaysia for providing their available software and robotic platform.

**Conflicts of Interest:** The authors declare no competing interest.

**Abbreviations**

The following abbreviations are used in this manuscript:

| | |
|---|---|
| ABC | artificial bee colony |
| ACO | ant colony optimization |
| AI | artificial intelligence |
| BA | bees algorithm |
| BFO | bacterial foraging optimization |
| BM | Brownian motion |
| FA | firefly algorithm |
| GES | group explosion strategy |
| GSO | glow-worm swarm optimization |
| IGES | improved group explosion strategy |
| LF | Lévy flight |
| MOPSO | multi-objective particle swarm |
| PSO | particle swarm optimization |
| RbRDPSO | repulsion-based robotic Darwinian particle swarm optimization |
| RDPSO | robotic Darwinian particle swarm optimization |
| RFID | radio frequency identification swarm optimization |
| RPSO | robotic particle swarm optimization |
| RW | random walk |
| SI | swarm intelligence |
| SLR | systematic literature review |
| SR | swarm robotics |
| TFS | triangle formation search |
| VREP | virtual robot experimental platform |

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
