# Peer review of "Systematic Literature Review of Swarm Robotics Strategies Applied to Target Search Problem with Environment Constraints"

_applsci, doi:10.3390/app11052383_

Round 1
Reviewer 1 Report
Dear authors,
I find the systematic review paper a very well structured paper and nicely addressed. However, for a review manuscript you are missing many many references to include in table 2 and 3.
It is not very understandable how the highest citation rank of the 12 most cited papers has 17 citations. The authors are missing tons of SI papers with many more citations.
I also see some problems on the conferences and journals selected:
What about swarm intelligence, ICRA, IROS, ALIFE, Swarm and evolutionary computation,........
You should review your sources of information and provide higher impact journal and conference papers
Author Response
We thank the Reviewer for his/her assessment of the presentation of our work.
Please see the attachment

Reviewer 2 Report
This paper aims to provide a systematic literature review of swarm robotics strategies applied to target search problems.
The reviewed papers were extracted from two scientific databases (IEEEXplore Digital Library and Springer) by applying a cited methodology for filtering results based on some research questions and exclusion/quality criteria.
The idea of this systematic review could be interesting, but it shows some shortcomings and relevant works omissions. Moreover, the structure of the paper is very similar to the cited Survey [21] in terms of both structure and covered topics.
Nevertheless, the most serious lacuna in the literature is the lack of significant studies citations of important authors like E. Bonabeau, M. Dorigo, G. Theraulaz, A. Martinoli, M. Gambardella, who made SR history. According to this reviewer's opinion, this review should enumerate more articles since it should represent a wide and more complete overview of this topic.
In details, I have the following specific comments:
- line 49-50: erroneous citations
ref. [7] refers to Bees Algorithm (BA)
ref. [8] refers to Artificial Bee Colony Optimization (ABC)
A Dorigo's paper ref. is mandatory for Ant Colony Optimization (ACO)
- Please carefully check to improve some incorrect or missing references’ information.
- Figure 1 is laid out vertically but a horizontal layout is preferred.
- Figure 3 is this useful?
- A minor revision is needed to improve the quality of English, more specifically, some sentences are no so clear, verbose and repetitive.
For example, line 433: “Some strategies combining two or more strategies to eliminate the limitation of one’s strategy by utilizing another strategy advantage.”
- Sincerely, my major concerns are directed at the lack of significant journal papers among the analysed ones. Several doubts arose from this lack. Please, justify this aspect better.
On the whole, the paper needs major refurbishments, including the introduction of more significant research papers published in International journals, perhaps considering further databases.
REF.:
[21] M. Senanayake, I. Senthooran, J. C. Barca, H. Chung, J. Kamruzzaman, and M. Murshed, “Search and tracking algorithms for swarms of robots: A survey,” Rob. Auton. Syst., vol. 75, pp. 422–434, 2016, DOI: 10.1016/j.robot.2015.08.010.
Author Response

(The authors gave the same response as above.)

Reviewer 3 Report
- Include an acronym section (before bibliography) instead table 1.
- I missed some results of a classic problem using different SR algorithms.
It would be great to include, if possible, such comparison with a predefined problem.
- When you mention "SR strategies and how there are implemented on the target search problem" (section 4.4.1 line 273) ... where are the results/comparison of each algorithm ???
- Any GitHub repo with code?
Author Response
We thank the Reviewer for his/her assessment of the presentation of our work. Next, we address each point raised by the reviewer.
Please see the attachment.

Round 2
Reviewer 1 Report
Accept in present form
Author Response
We thank the Reviewer for his/her assessment of the presentation of our work. Next, we address each point raised by the reviewer in the following order:
Point 1: Accept in present form.
Author response: Thank you very much for the consideration, and acceptance.
Reviewer 2 Report
In general, the paper was improved and now is more clear. The authors revised the manuscript according to the reviewers’ comments including additional details and the analysis of further papers.
However, I still have some remaining comments and doubts about it, listed below:
- some Figures (1, 4, 5) and Tables (1,2,3) exceed the margins of the document. Please fix the layout.
- line 139: missing comma between IEEE Xplore and ScienceDirect --> (Springer, IEEE Xplore ScienceDirect, and MDPI)
- After the first revision process the authors have introduced two more databases (ScienceDirect and MDPI) with the result to increase from 19 to 25 the final outcomes of the selected papers. Nevertheless, the authors declare the survey had taken place in October 2020.
How is it possible? If it is not a typo and all databases were already included during the original query processing, why the authors decided to focus only on Springer and IEEE Xplore, given the small number of papers? Please, clarified this doubt.
- The authors introduced a reference to a GitHub repo in the Data Availability Statement section as a response for Reviewer 3’s question. The authors declare: “Some codes for the initial study of the authors in this domain can be retrieved in the GitHub repository (https://github.com/xrl2408/E2RPSO) [63]” but it is not clearly discernible from the authors' names that do not match. What is the benefit to introduce this citation, since the publication seems neither from the same authors nor cited in any manuscript sections? Please, justify this choice.
In this context, a synthetic review on the availability of open-source repos (yes with links or not) for ALL the papers selected would have been rather useful.
Author Response
Many thanks for the constructive comments here.
Kindly see the attachment for our responses on each point raised by the reviewer.
